# ER exit sites mediated by the COPII adaptor sec24D selectively recruit lipid raft-preferring proteins for rapid ER export

Ivan Castello-Serrano [1,2] ✉, Shikha Dagar [1,3], Rossana Ippolito[1,2], Kandice R. Levental [1] ✉ & Ilya Levental [1] ✉

The determinants of sub-cellular trafficking for many membrane proteins are poorly understood. Lipid-driven membrane nanodomains known as lipid rafts have been widely implicated in post-Golgi traffic, but their involvement in protein sorting in the endoplasmic reticulum has not been widely considered. To assess the role of membrane domains in the early secretory pathway, we use the Retention Using Selective Hooks system to synchronize and quantitatively assess trafficking rates and destinations of model proteins with tunable raft affinities. We find that raft-preferring constructs exit the ER faster than raft-excluded and have distinct preferences for ER exit sites marked by specific isoforms of sec24 cargo adaptors. Namely, raft-excluded cargo localizes to sec24A-positive sites while raft-preferring cargo localizes to sec24D ERES, dependent on p24-family cargo adapters TMED2/10. Finally, sec24D, but not sec24A, ERES accumulate a fluorescent cholesterol analog. These observations suggest that association with raft-like domains affects protein export from the ER.

Most eukaryotic membrane proteins originate at the endoplasmic reticulum (ER), where they are folded and integrated into the lipid bilayer. Proteins destined for other organelles must then exit the ER, a process mediated by an assembly of sec24 family cargo adapters and sec31/13 coat proteins (among other factors) known as the Coat Protein Complex II, or COPII[1,2]. These large assemblies sort both soluble and transmembrane proteins[1,3,4] into specialized domains called ER exit sites (ERES), from which vesicular and tubular[5] carriers transfer cargo to the sorting hub of the Golgi Apparatus. While the general mechanisms of COPII function have been widely explored, how cargo is selected for export and how secretion is regulated are less understood.

Lipids and proteins can be laterally sorted by membrane nanodomains known as lipid rafts[6]. Rafts can arise due to the intrinsic capacity of biomembranes to laterally separate into coexisting ordered and disordered domains[6,7]. Due to their rapid dynamics and small size, it remains difficult to directly visualize raft domains in living cells—with some notable exceptions[8–10]—however, accumulating evidence

has established important roles for lipid-mediated membrane organization[6,11]. One of these roles, indeed the original function for which lipid rafts were proposed[12], is membrane sorting in secretory traffic[13], including plasma membrane (PM) homeostasis[14], endocytic sorting[14,15], and kinetics of Golgi efflux[16]. A significant advance allowing insights into the functions of raft association is the direct visualization of ordered lipid phases in isolated PMs known as Giant Plasma Membrane Vesicles (GPMVs)[6,17–19]. GPMVs phase separate into coexisting ordered and disordered domains that laterally sort membrane components according to their preference for raft-like environments[20–22]. The relatively ordered "raft" phase enriches saturated lipids, glycolipids[17], GPI-anchored proteins[23], and selected transmembrane domains (TMDs)[24], while unsaturated phospholipids and many larger transmembrane proteins are excluded[25]. Thus, GPMVs provide a robust tool to quantify the intrinsic affinity of membrane components for raft-like domains in biomembranes. Importantly, raft preference for some proteins is tightly correlated with subcellular localization, with raft association being necessary and sufficient for PM localization[15].

[1]Department of Molecular Physiology and Biological Physics, Center for Membrane and Cell Physiology, University of Virginia, Charlottesville, VA, USA. [2]Present address: IRCCS Ospedale San Raffaele, Milan, Italy. [3]Present address: Department of Microbiology and Immunology, University of Illinois Chicago, Chicago, IL, USA. ✉e-mail: castelloserrano.ivan@hsr.it; krl6c@virginia.edu; il2sy@virginia.edu

While these and other[13] observations suggest that lipid-driven membrane nanodomains are involved in endocytosis and recycling, the role of rafts in the early secretory pathway remains controversial and poorly understood. This controversy persists because rafts are not easily visualized, most methods for raft disruption are non-specific[26], and classical protocols for measuring raft association are artifact-prone, non-quantitative, and difficult to interpret[27]. To directly interrogate the influence of raft affinity in secretory traffic, we recently constructed a panel of minimal protein probes with defined preferences for membrane domains and measured their trafficking using RUSH, a robust tool for synchronized secretory traffic[21]. Here, we investigate how cargo protein raft affinity affects their ER efflux and recognition by the inner layer of the COPII machinery responsible for ER export. We find that raft-preferring TMD probes have faster ER efflux than parallel raft-excluded ones, suggesting that raft preference affects protein sorting and trafficking in the ER. This difference in ER efflux rates correlates with the distinct localization of raft-preferring versus raft-excluded TMDs and their association with distinct COPII components. Namely, raft-preferring TMDs are more likely to transit through ERES marked by sec24D, whereas nonraft TMDs transit sec24A-positive ERES. The selective association of sec24D with raft-associated TMDs is mediated by the TMED2/TMED10 complex of p24-family cargo adapters. Consistent with the apparent selectivity of sec24D ERES for raft-associated cargo, these sites also accumulate fluorescent cholesterol analogs. Finally, inhibition of raft-forming lipids disrupts sec24D ERES and retards ER exit kinetics of raft-preferring proteins. These observations suggest that lipid-driven domains are involved in functional cargo sorting at ERES.

## Results

### Raft-preferring LAT exits ER faster than nonraft mutant

In previous work, we interrogated raft-dependent protein trafficking kinetics in the secretory pathway using the single-pass transmembrane adapter Linker for Activation of T-cells (LAT) as a model PM-targeted cargo[14–16]. The preference of proteins and lipids for raft domains can be quantified directly by measuring their partitioning between ordered and disordered phases in isolated GPMVs, as shown in Fig. 1A and Supplementary Fig. 1A. As previously reported[14–16,23,24], LAT prefers the raft phase of GPMVs, which can be identified by its exclusion of the unsaturated, nonraft lipid marker F-DiO (Fig. 1A, top). In contrast, replacing the native LAT TMD with a 22-Leu TMD (i.e., LAT-allL) leads to its exclusion from raft domains and a strong preference for the nonraft phase (Fig. 1A, bottom). This behavior is quantified through the raft partition coefficient ($K_{p,raft}$), i.e., the ratio of protein concentration in the raft versus nonraft phases (quantified via the background-subtracted intensities of their linked fluorophores) (Supplementary Fig. 1A).

To assess how raft partitioning affects the kinetics of secretory trafficking, we inserted LAT and LAT-allL into the RUSH system for synchronized release and tracking through the secretory system (Fig. 1B). This assay is based on the reversible interaction of a target protein fused to streptavidin-binding peptide (SBP) with a selective "hook"; here, streptavidin retained in the ER via the KDEL motif. This hook-SBP interaction retains proteins of interest (POIs) in the ER until introduction of biotin induces rapid, synchronous release of POIs from the ER, whose progress through the secretory system is then traced via fluorescent protein tags. HEK cells were co-transfected with LAT-EGFP and LAT-allL-RFP, which were simultaneously released from the RUSH hook by biotin (Fig. 1C). These constructs showed notably different ER exit kinetics, with raft-preferring LAT accumulating in the Golgi after 45 min and completely emptying from the ER after 90 min, whereas raft-excluded LAT-allL showed a strong ER signal at both time points. This observation was quantified by time-lapse microscopy of live cells to measure the fraction of protein remaining in the ER (Fig. 1D), which can be summarized as the "ER residence time" (time at 50% remaining

in ER), revealing that LAT exits the ER ~3-fold faster than LAT-allL (Fig. 1D, E). Importantly, both LAT and LAT-allL eventually reach their steady-state localization at the PM and endosomes (Supplementary Fig. 1B).

### Cargo raft association is related to the sec24 isoform colocalization

The distinct ER efflux rates for raft versus nonraft LAT constructs led us to hypothesize that they associate differently with ER trafficking machinery. We recently showed that LAT has a short cytosolic motif that mediates its rapid ER efflux, likely through engagement of COPII[16]. During assembly of the COPII coat, sec24 proteins are responsible for cargo recognition and recruitment to ERES[2,28]. Humans have four sec24 isoforms, with sec24A/B forming a sub-family (based on sequence similarity) separate from sec24C/D[28]. These isoforms have been previously implicated in the selective trafficking of certain cargo, with sec24C/D involved in ER efflux of GPI-anchored proteins[1]. Immunostaining against the four endogenous sec24 isoforms in both HEK and HeLa showed similar distributions for sec24A and B, which were notably distinct from those of sec24C and D (which were similar to each other) (Supplementary Figs. 2 and 4). Namely, sec24A/B puncta were distributed throughout the cytoplasm (Fig. 2A), while sec24C/D puncta were largely perinuclear, reminiscent of Golgi localization (Fig. 2B). To confirm and quantify this effect, we analyzed the distribution of sec24A or sec24D relative to 58K, a cis-Golgi marker. Across multiple experiments, sec24D distribution was strongly correlated with the Golgi marker, while sec24A was uncorrelated (Fig. 2C, Supplementary Fig. 3). Importantly, both sec24A and sec24D colocalized with a marker for COPII, supporting that these antibodies report the presence of discrete ERES (Fig. 2D–F). We further validated that the antibodies were specific for the sec24 isoforms by immunostaining cells expressing fluorescence-tagged sec24 constructs and observing that anti-sec24A stained the same puncta observed with sec24A-mCherry while anti-sec24D colocalized with sec24D-mCherry (but not vice versa) (Supplementary Fig. 5). Thus, sec24A and sec24D define distinct ERES with distinct localizations.

To determine which of these ERES were used by LAT variants, we used RUSH to synchronize their ERES residence. Fifteen minutes after biotin introduction, RUSH-synchronized LAT accumulated in bright ER puncta that colocalized with sec24D puncta (Fig. 3A, C). Nonraft LAT-allL also formed puncta, but these did not colocalize with sec24D-positive structures. Rather, LAT-allL puncta colocalized with sec24A (Fig. 3B, C), which did not colocalize with LAT (Fig. 3A, C). The same effect was also observable when LAT and LAT-allL were co-expressed in the same cells (Supplementary Fig. 6).

This selectivity of sec24D for raft-preferring cargo also held for other proteins: GPI-anchored EGFP (GPI) and tumor necrosis factor α (TNF-α)−both previously associated with raft domains[5,22,23] and confirmed here to enrich in the raft phase of GPMVs (Supplementary Fig. 1)−also colocalized with sec24D. In contrast, transferrin receptor (TfR), a classical nonraft protein excluded from GPMV raft phases[18] (Supplementary Fig. 1), colocalized strongly only with sec24A puncta. Thus, raft-preferring constructs were enriched into distinct ERES from nonraft proteins, marked by different isoforms of sec24.

To investigate molecular mechanisms underlying these effects, we focused on the p24 family of early secretory pathway adapter proteins, which are involved in the recruitment of cargo to COPII[28,29] and COPI coat complexes[30]. Notably, the p24 family members TMED2 and TMED10 play a key role in recruiting GPI-anchored proteins to sec24C/D ERES[28,31], implicating them in ER efflux of raft-associated proteins. More recently, TMED2/10 (which are believed to form heteromeric complexes) were implicated in lipid transport required for the formation of raft-like membrane domains[32]. Thus, we investigated whether these proteins are involved in the selective recruitment of raft-associated proteins to sec24D ERES. Using siRNA, we knocked

 

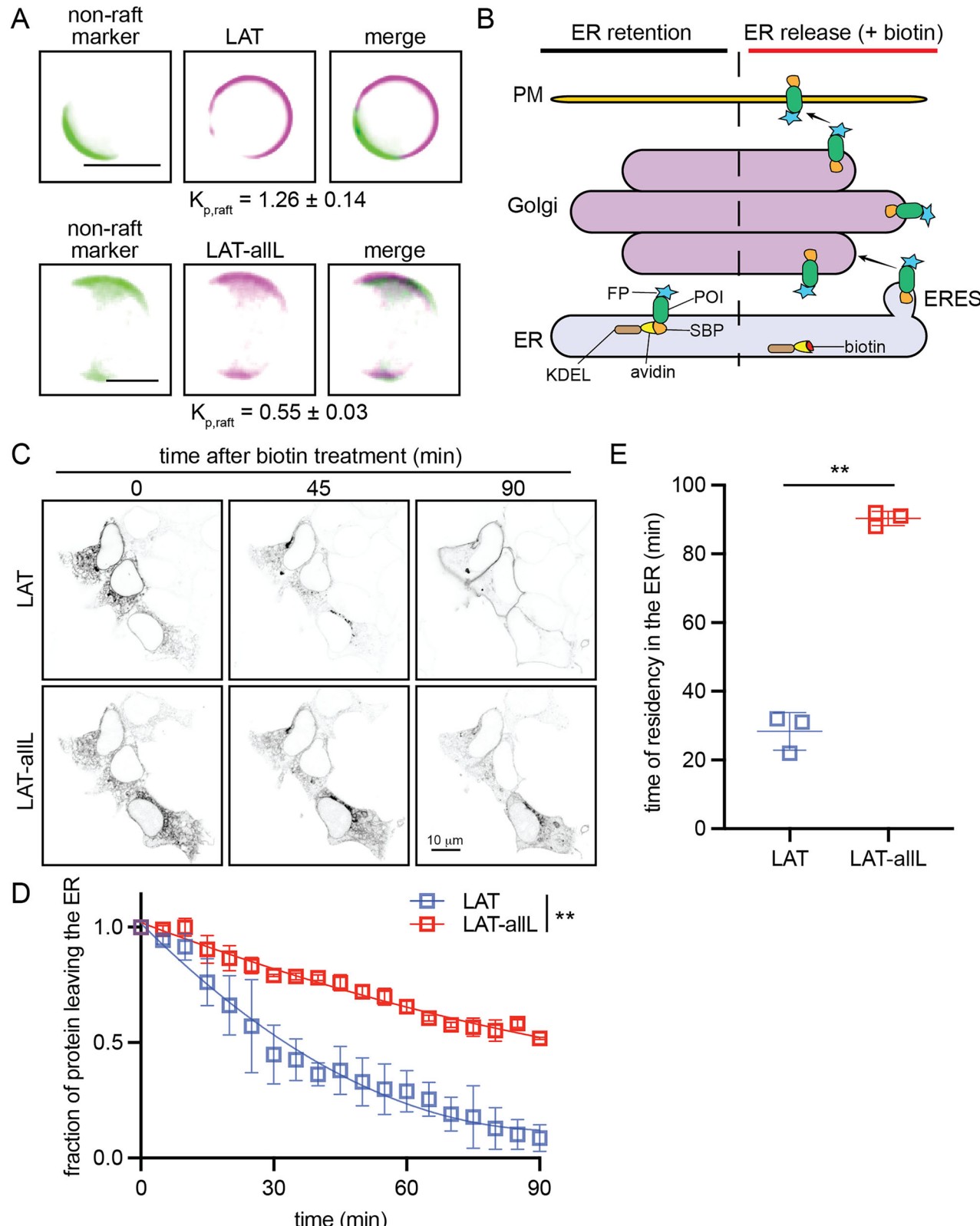

down the expression of both proteins (Fig. 4A) and confirmed the previously reported effects on GPI-AP localization and ER export[28]. Namely, ER efflux of GPI-EGFP was greatly attenuated, as quantified by the fraction of GPI-EGFP signal in the Golgi either 15 or 30 min after its release from the ER with biotin (Fig. 4B, D); TMED2/10 KD reduced the Golgi fraction of GPI-EGFP from >60% to <20% at 15 min after its release from the ER (Fig. 4D). Concomitantly, we observed a significant delay

in the association of GPI-EGFP puncta with sec24D puncta (Fig. 4G). Similar, though quantitatively smaller, effects were observed for LAT: knockdown of TMED2/10 reduced its Golgi localization in the first 30 min after release of ER retention (Fig. 4C, E) and also reduced its association with sec24D puncta (Fig. 4H). Neither of these effects was observed for LAT-allL (Fig. 4F, I). Thus, TMED2/10 are adapters for the selectivity of sec24D ERES for raft-associated proteins.

**Fig. 1 | Raft-preferring LAT is exported faster from ER than raft-excluded LAT-allL. A** Images of GPMVs labeled with F-DiO (green) to indicate the nonraft phase prepared from cells transfected with either (top) raft-preferring LAT or (bottom) raft-excluded variant LAT-allL (magenta). Scale bar is 10 μm. **B** Schematic of the RUSH synchronous trafficking system. A transmembrane protein of interest (POI) is retained in the ER via the interaction of a hook (KDEL ER retention signal fused to streptavidin) and an SBP tag on the POI; introduction of biotin releases the POI. **C** Timelapse confocal imaging of cells co-transfected with RUSH-synchronized LAT and LAT-allL. Raft-preferring LAT exits the ER faster, with complete ER exit at 90 min, whereas LAT-allL is delayed. Scale bar = 5 μm. **D** LAT and LAT-allL ER exit

kinetics were monitored by live imaging. The fraction of construct in the ER was quantified by using a mask of the $t = 0$ min signal, wherein all constructs are exclusively ER resident. These curves were fitted with single exponentials to calculate the "residence time" in the ER (time with 50% protein remaining in ER). **E** Quantification of ER residence time showing that LAT exits the ER 3-fold faster than LAT-allL. **D** Points represent means ± S.D. of three independent experiments with >5 cells each. The result of the two-way ANOVA test is shown. **E** Points represent individual means of three independent experiments with >5 cells each. Lines are mean ± S.D. Result of paired t-test shown. **p < 0.01.

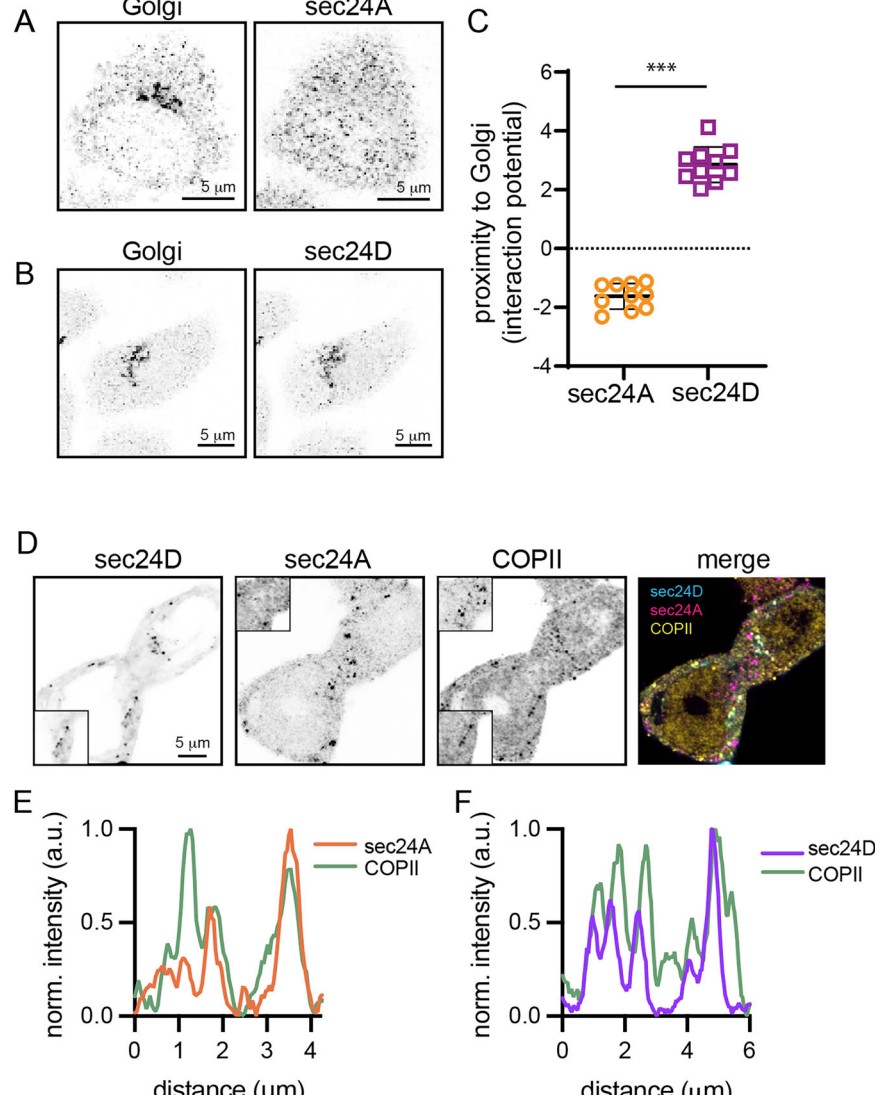

**Fig. 2 | sec24 isoforms localize to COPII sites but show different subcellular distributions. A** HEK cells immunostained with anti-58K, a Golgi marker (left), and sec24A (right), showing no enrichment of sec24A in the perinuclear area containing the Golgi marker. **B** Similar staining against 58 K (left) and sec24D (right) showed clear enrichment of sec24D near the Golgi. **C** Potential interaction values for either sec24A (orange circles) or sec24D (purple squares) with the Golgi marker. Symbols represent values from 10 individual cells from a single experiment, representative

of two independent repeats. Shown is the mean ± S.D. and the result of an unpaired t-test of one experiment. **D** HEK cells transiently co-expressing sec24D-EGFP and sec24A-mCherry were stained with a COPII antibody. **E, F** Line scans show that both sec24 isoforms independently colocalize with COPII. Insets in (**D**) show region of quantification (top inset is sec24A-COPII colocalization, bottom inset is sec24D-COPII colocalization). Scale bars are 5 μm. ***p < 0.001.

## Fluorescent cholesterol accumulates in sec24D, but not sec24A, puncta

Despite cholesterol being synthesized in the ER, its concentration in the ER membrane is low relative to many other organelles[33,34]. However, it was recently reported that a fluorescent cholesterol analog (TF-chol) accumulates in ER-associated puncta[5]. We

reproduced this observation by staining live cells with TF-chol (Supplementary Fig. 7), though we note that TF-chol distribution was strongly dependent on time and labeling concentration, potentially representing flux of excess cholesterol through various cell membranes[35–37]. These TF-chol puncta colocalized strongly with sec24D-positive structures, but much less with sec24A (Fig. 5).

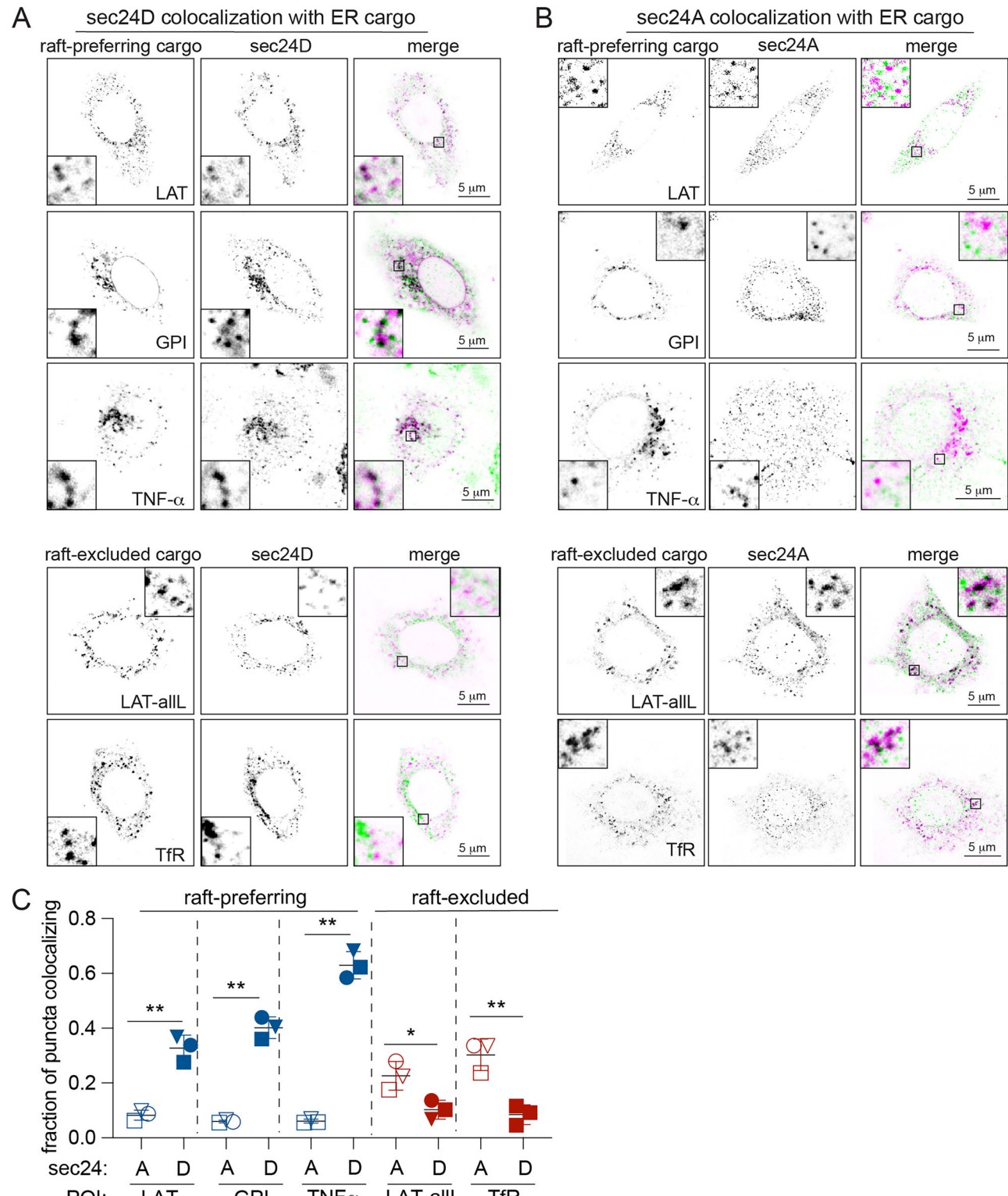

**Fig. 3 | sec24 isoforms colocalize with cargo proteins based on their raft affinity. A** Confocal images of cells expressing RUSH-synchronized cargo proteins 15 min after biotin introduction (left) and immunostained against sec24D (middle). In merge, green = sec24D, magenta = cargo (marked by FP), black = colocalization. Raft-preferring proteins selectively colocalize with sec24D puncta. **B** Same cargo proteins and conditions as (**A**), but immunostained against sec24A. Nonraft-preferring proteins selectively colocalize with sec24A puncta. **C** Quantification of protein puncta colocalizing with sec24 puncta. Blue represents raft-preferring proteins; red for raft-excluded proteins. Symbols correspond to means ± S.D. of three individual experiments with >10 cells each. Empty symbols represent colocalization with sec24A, solid symbols with sec24D. Results of the Welch's $t$-test are shown. $*p < 0.05$, $**p < 0.01$. Scale bars = 5 µm.

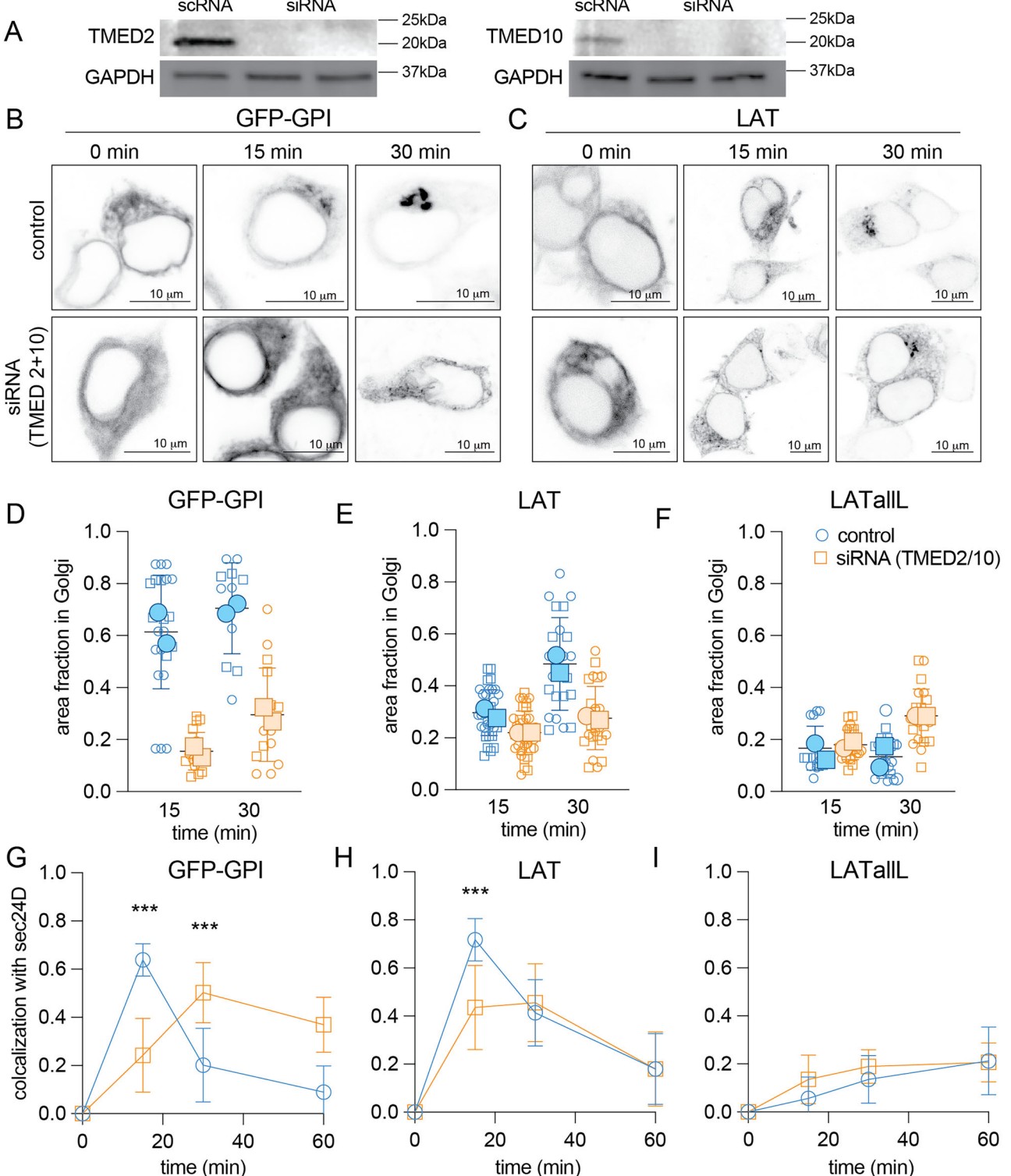

**Fig. 4 | TMED2/10 mediates raft-associated protein recruitment to sec24D.**
**A** Western blotting reveals knockdown of protein expression of both TMED2 and TMED10 by siRNA. Controls are scrambled siRNA sequences. Shown is a representative Western blot of two independent repeats with two independent KD experiments in each. Full uncropped Western blot shown in Supplementary Fig. S8. **B**, **C** Confocal images of cells expressing RUSH-synchronized cargo proteins at various time points after biotin introduction. TMED2/10 knockdown leads to more ER retention of **B** GPI-EGFP and **C** LAT. **D**–**F** Quantification of Golgi accumulation (via anti-58K mask) of RUSH constructs 15 and 30 min after release of ER retention. TMED2/10 knockdown reduces Golgi accumulation of **D** GPI-EGFP and **E** LAT, but not **F** LAT-allL. Small symbols represent individual cells; large symbols represent means of two independent experiments. **G**–**I** Quantification of temporal colocalization of RUSH cargo puncta with sec24D puncta. TMED2/10 knockdown reduces colocalization with sec24D of **G** GPI-EGFP and **H** LAT, but not **I** LAT-allL. Symbols represent means ± S.D. of two independent experiments. Results of unpaired *t*-tests with multiple comparisons are shown. ***$p < 0.001$. Scale bars = 10 μm.

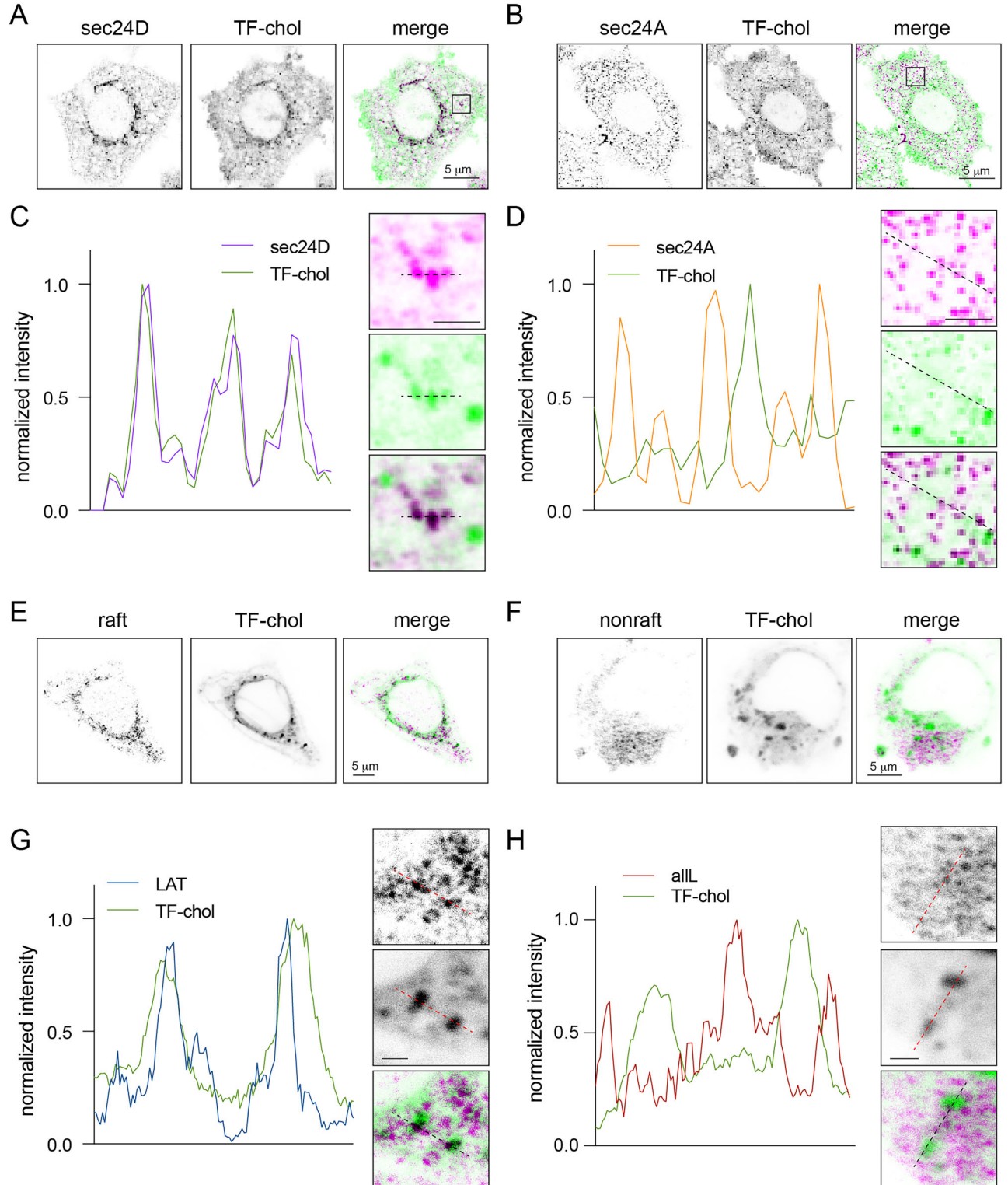

**Fig. 5 | TF-chol accumulates in sec24D, but not sec24A, puncta. A**, **B** Cells were stained for 15 min with TF-Chol (green), then imaged after 2 h at 37 °C. TF-Chol accumulates in intracellular puncta, which colocalize with sec24D but not sec24A puncta (magenta). In merge, black corresponds to colocalization. **C**, **D** 4-fold zoom of the boxed region in (**A**, **B**). Line scans showing TF-chol co-enrichment with sec24D but not **D** sec24A. **E**, **F** Similar TF-Chol staining in cells expressing RUSH LAT or LAT-allL 15 min after biotin treatment to release the constructs from the ER to ERES. LAT accumulates in TF-Chol-rich puncta, whereas LAT-allL does not. **G**, **H** 4-fold zoom of the boxed region in (**E**, **F**). Line scans showing TF-chol co-enrichment with LAT but not **H** LAT-allL. Images and line scans are representative of three independent repeats with >5 cells per experiment. Scale bars = 5 μm, inset scale = 1 μm.

Intensity profiles of magnified regions show clear overlap of TF-chol with sec24D (Fig. 5A), with less correlation with sec24A (Fig. 5B). Thus, consistent with the preference of sec24 isoforms for raft-associated proteins, sec24D puncta may be relatively enriched in cholesterol.

Consistently, we also observed that these cholesterol-enriched ER puncta colocalized with raft-TMD puncta produced 15 min after their release from the ER (Fig. 5E, G), but not with those of nonraft LAT-allL (Fig. 5F, H).

## Inhibition of raft lipid synthesis disrupts sec24D puncta and slows LAT ER exit

Since sec24D puncta appeared to enrich cholesterol and raft-associated membrane proteins, we hypothesized that raft lipids would be important for the formation and function of these structures. To test this hypothesis, we treated cells with inhibitors of key raft-forming lipids: myriocin (M) and zaragozic acid (ZA) to inhibit sphingolipid and cholesterol synthesis, respectively[16,38,39]. These treatments were previously shown to destabilize raft phases and suppress raft-dependent signaling and trafficking[16,38,39]. sec24D puncta were notably less pronounced after M-ZA treatment, in exchange for more cytosolic staining, while sec24A puncta were not affected (Fig. 5A). We quantified this effect by measuring sec24 intensity in puncta relative to the cytosol (Fig. 5B), demonstrating a highly significant and time-dependent reduction in the punctate morphology of sec24D but not sec24A. Thus, inhibition of raft lipids selectively disrupts sec24D puncta. To study the functional impacts of these perturbations, we measured how M-ZA treatment affected ER efflux kinetics of LAT and nonraft LAT-allL. In untreated cells, raft-preferring LAT had faster ER efflux than nonraft LAT-allL (Fig. 5C–E and also Fig. 1). M-ZA treatment significantly slowed ER efflux of LAT but did not affect LAT-allL. Notably, LAT and LAT-all ER exit kinetics were indistinguishable after M-ZA treatment (Fig. 5D, E), revealing that inhibition of raft lipid synthesis abolishes the faster ER efflux of raft-associated LAT. It is important to note that the effects of these inhibitors are not restricted to the ER, as both cholesterol and sphingolipids distribute broadly and enrich in the PM after their synthesis, thus our observations do not necessarily implicate changes in ERES lipid composition.

## Discussion

The involvement of lipid-driven domains in subcellular sorting of membrane lipids and proteins has been widely studied in endocytosis and late secretory traffic[13]. Here, we explored the role of raft partitioning in ER efflux by combining engineered proteins, direct measurements of raft affinity in GPMVs, and RUSH to achieve controlled, synchronized cargo traffic. We find that raft-associated LAT has faster ER efflux than a raft-excluded mutant (Fig. 1). These kinetics are dependent on the synthesis of raft-forming lipids, cholesterol, and sphingolipids (Fig. 6). The differences in ER efflux are correlated with the differential sorting of transmembrane cargo and cholesterol with distinct sec24 isoforms (Fig. 3). These observations imply a role for lipid-mediated domains in the organization and function of the early secretory system.

This implication is surprising in light of the ER's relatively low levels of cholesterol[33,40,41], which is essential for the formation of ordered lipid domains[7,42]. Similarly, complex sphingolipids—including sphingomyelin and glycosphingolipids—are synthesized in the Golgi and therefore unlikely to be present at significant abundances in the ER[43]. However, membranes with cholesterol concentrations at or even below 10 mol% can phase separate[44,45], even without considering the possible influence of local differences in lipid compositions (e.g., due to local production or lateral confinements) or the effects of protein assembly on membrane organization[10]. Indeed, such multimeric protein assemblies underlie the structure of ERES, suggesting the possibility that protein-protein interactions induce the clustering of raft-preferring transmembrane proteins, which in turn recruit raft-preferring lipids like cholesterol. We suggest that the p24 family adapters, in our case TMED2/10, may be involved in such assemblies, perhaps through their direct interactions with GPI-anchored proteins[28]. Such effects may also be responsible for the cholesterol-enriched ER structures that were recently reported[5] and confirmed here (Fig. 5). Weigel et al. demonstrated that these structures are ERES and accumulate not only cholesterol analogs, but also the raft-preferring protein TNFα, while previous literature[46] and our data (Fig. 6C–E) reveal that cholesterol depletion disrupts ER-to-Golgi transport.

The enrichment of a fluorescent cholesterol analog (TF-cholesterol) in ER structures[5] (Fig. 5) is surprising, since the majority of native cellular cholesterol is in the PM and endosomes[33,40]. First, it is important to emphasize that fluorescent lipids are imperfect analogs of their native counterparts, though TF-Chol retains the ordered phase affinity of native cholesterol better than most[47,48]. We analyzed staining at various concentrations and time points, noting that while TF-Chol efficiently stained ER structures under some conditions, lower concentrations and shorter labeling times showed no ER staining but rather PM accumulation (Supplementary Fig. 7). We suspect that these distributions reflect aspects of subcellular cholesterol traffic, especially under cholesterol over-loading, which may allow visualization of cholesterol-rich ERES structures that would not be observable in steady-state.

Our major finding is that ERES defined by sec24A versus sec24D select cargo based on their lipid raft partitioning, with sec24D recruiting raft-preferring proteins and vice versa (Fig. 3). It is notable that the differentially recruited cargo constructs have identical cytosolic and lumenal features, including LAT's putative COPII interaction motif[16], thus these results cannot be explained by the distinct specificities of the sec24 isoforms for various cargo motifs[49]. We find that these two types of ERES have distinct subcellular distributions, with sec24D sites being more perinuclear and Golgi-proximal (Fig. 2). Inhibition of raft-forming lipids slows down, but does not block, ER export of LAT, coincident with disrupted assembly of sec24D (Fig. 6A, B). While our observations (Fig. 2) imply a dichotomy of sec24 isoforms (A/B and C/D) previously established through sequence homology[50], this classification remains unsettled[49,51]. Some studies support this grouping based on genetic rescue experiments[52], while others suggest alternative groupings[53]. Though we did not study sec24B and C, we predict that the raft selectivity we reported for sec24D would be retained for sec24C and vice versa, which would be consistent with reports that sec24C/D are selectively responsible for ER exit of GPI-anchored proteins[28] (which have been extensively linked with raft domains[23]).

It was recently demonstrated that ERES can segregate cargo proteins based on their physical characteristics (i.e., molecular size)[54]. Our observations suggest that a related principle may apply to TMDs, with features that impart raft affinity[24] leading to enrichment of cargo in sec24D-rich ERES and subsequent fast ER efflux. Functionally, a mechanism for rapidly exporting raft-preferring TMDs may be important for avoiding ER membrane stress[55,56] that could be induced by accumulation of raft-forming saturated lipids[57] or mis-fitting TMDs[58]. Altogether, these observations suggest that selective lipid-driven domains can participate in organizing cargo and trafficking machinery in ERES for efficient export of PM-directed transmembrane proteins.

## Methods

### Cell culture and treatment

HEK-293NT (HEK) and HeLa cells were purchased from ATCC and cultured in medium containing 89% Dulbecco's Modified Essential Medium high glucose, 10% FCS, and 1% penicillin/streptomycin at 37 °C in humidified 5% $CO_2$. Transfection was done by Lipofectamine 3000 using the manufacturer's protocols. 4–6 h after transfection, cells were washed with PBS and then incubated with serum-free medium overnight. To synchronize the cells, 1 h before biotin addition, the cells were treated with full-serum medium. Lipid synthesis inhibitors (25 μM myriocin and 5 μM zaragozic acid) were added to the medium 2 days before transfection and during the whole experiment. TopFluor-cholesterol (Avanti, cat# 810255) was prepared following the instructions in Weigel et al.[5]. To monitor TF-chol subcellular distribution, HeLa cells were incubated with TF-chol for 15 min, then chased in label-free medium for various times, allowing TF-cholesterol to be trafficked prior to imaging experiments. We found that the optimal

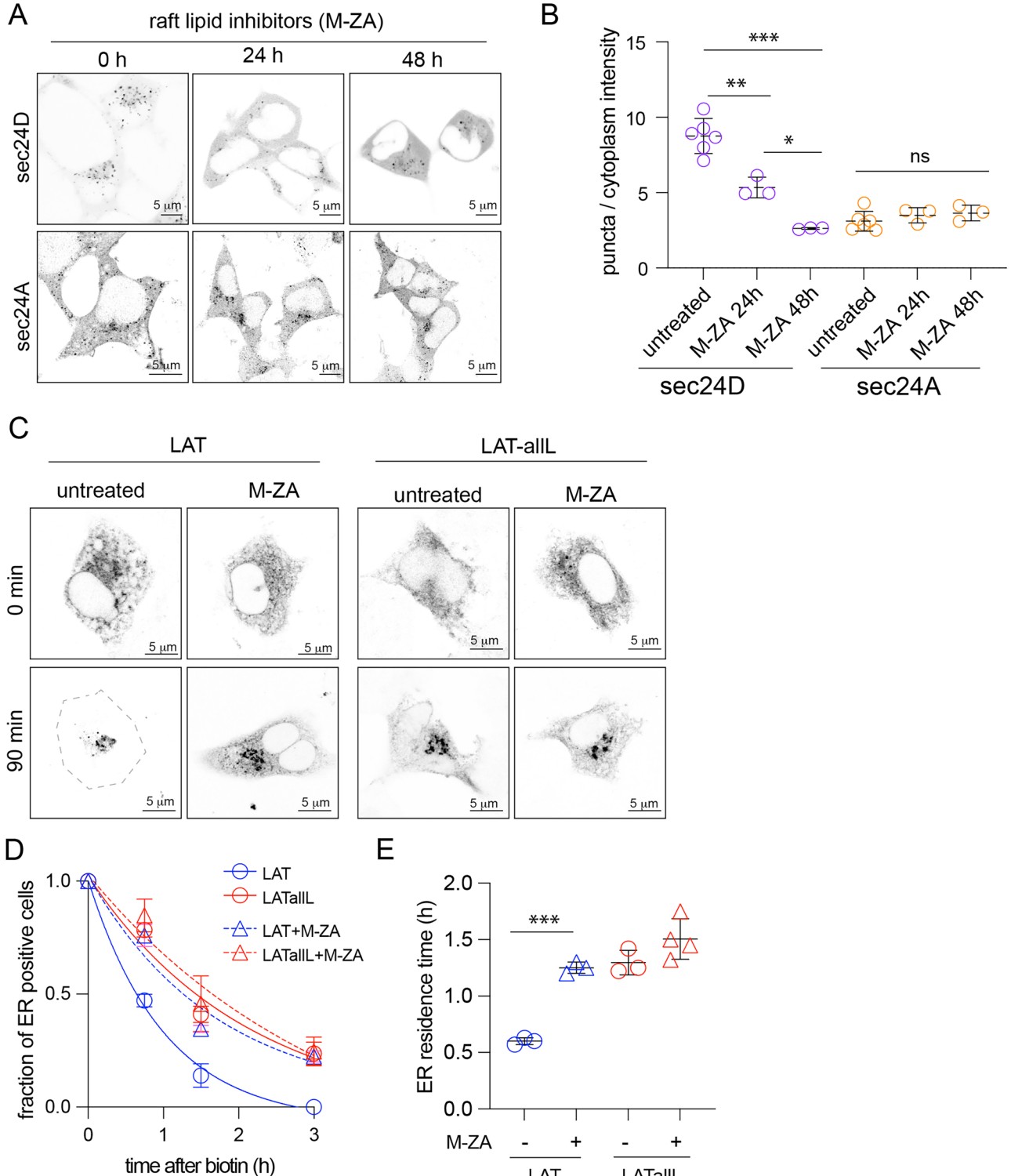

**Fig. 6 | sec24D assembly and fast LAT ER efflux are disrupted by raft lipid inhibition. A** Confocal images of cells transfected with either sec24A or sec24D before and after treatment with myriocin and zaragozic acid (M-ZA) to inhibit the synthesis of raft-forming lipids. The intensity of sec24D puncta is reduced in favor of cytosolic localization, whereas sec24A is not affected. **B** Quantification of puncta relative to cytosol intensities. Treatment with M-ZA reduces the intensity of sec24D puncta with no significant effect on sec24A. Shown are the mean ± S.D. of three independent experiments (>25 cells each) and results from the Welch's *t*-test. **C** Confocal images of representative LAT RUSH experiments show that ER efflux is retarded by M-ZA treatment while LAT-allL is unaffected. **D** Quantification of cells with ER staining over time. LAT export is retarded by M-ZA treatment, becoming similar to LAT-allL. Symbols represent mean ± S.D. of three independent experiments with >5 cells each. Shown is a one-phase association fit. **E** ER residence time calculated from fitting curves in (**D**); LAT's faster efflux than LAT-allL is eliminated by M-ZA treatment. Shown are the mean ± S.D. of three independent experiments (>5 cells each) and results from the Welch's *t*-test. Scale bars are 5 μm. *$p < 0.05$, **$p < 0.01$, ***$p < 0.001$.

concentration and timing for ER puncta staining was using 1 µM for 2 h (Supplementary Fig. 7).

### Plasmids and mutations

All RUSH protein constructs were based on the bicistronic LAT and LAT-allL (full length) RUSH backbone previously described[16]. For LAT-allL, the amino acid sequence of the LAT TMD (NH2-MEEAILVPCVL GLLLLPILAMLMALCVHC) was replaced with NH2-MEELLLLLLLLLLL LLLLLLLLLCVHC. Constructs purchased from Addgene: GPI RUSH version (#65294 (EGFP), #65295 (Cherry)), RUSH version of full-length TNFa (#65279), sec24D (#32677 (mCherry), #32678 (EGFP)). RUSH version of full-length TfR was obtained from Frank Perez's lab (Institut Curie), sec24A-mCherry was obtained from Susan Ferro-Novick's (UC San Diego) lab.

### RUSH expression and chase

In brief, cells transfected with RUSH plasmids were incubated overnight at 37 °C, then treated with 100 µM biotin (Sigma-Aldrich) to release cargo proteins. Cells were incubated at 37 °C for various chase times and either directly imaged live or fixed with PFA as described below.

### Imaging and quantification

Unless specified, imaging was performed on a confocal microscope using three-channel acquisition (green = ex:488/em:509, red = ex:584/em:609, far-red = ex:584/em:610) in a sequential manner. When possible, cells were imaged live; for immunofluorescence, cells were fixed with 4% paraformaldehyde (PFA) for 10 min at 23 °C. anti-58k (1:500 mouse, Abcam ab27043) was used as a cis-Golgi marker. Antibodies against sec24 isoforms were Sigma-Aldrich sec24A HPA038901, sec24B HPA038181, sec24C HPA040196, sec24D HPA053486, sec24A Santa Cruz sc-517155, sec24D ThermoFisher 67409-1-IG. The polyclonal COPII antibody was ThermoFisher PA1-069A.

Immunostaining was performed with secondary antibodies conjugated with Alexa fluorophore dyes. Acquired images were quantified using Fiji. Coloc2 measurements were used to calculate Pearson's coefficient. Puncta colocalization was calculated by identifying puncta using the Fiji thresholding algorithm, then manually counting the LAT/LAT-allL puncta overlapping with sec24A/sec24D puncta by the total number of LAT/LAT-allL puncta. For Fig. 6D, "Fraction of ER-positive cells" was calculated by manually scoring cells with ER signal above background at various time points; for Fig. 6E, "ER residence time" was calculated by fitting the curves in Fig. 6D to a single-exponential decay and deriving the time at which 50% of cells were ER positive. To calculate the fraction of construct in the Golgi (Fig. 4), a Golgi mask was created by immunostaining fixed cells against the Golgi marker anti-58K, then comparing the fluorescence signal from the Golgi mask to the whole cell.

For the time-lapse experiment in Fig. 1D, E, we used confocal imaging of live cells expressing RUSH constructs of either LAT or LAT-allL (tagged with fluorescent proteins). At each time point, the fluorescent protein intensity in the ER was measured by applying a mask generated from the $t = 0$ time point (i.e., without biotin), at which all signals for all constructs are in the ER. This fluorescence in the ER was then divided by the fluorescence in the cell as a whole to calculate the fraction of target protein in the ER. This "fraction remaining in the ER" was then quantified in images taken every 5 min after biotin introduction.

### siRNA knockdown

Expression of TMED2 and TMED10 was silenced using siRNAs obtained from Qiagen (siTMED10: AGGAGTTTATCCTTTCCGTAA; siTMED2: TAGGTCCTTCCAGGAACTCAA) and using scrambled versions of the same RNAs as controls (scTMED10: GGTTTCAGCTCGTTACTAAAT; scTMED2: GCTGATCAAACCCGTATGACT). Transfection of RNA constructs was done by Lipofectamine RNAiMAX following the

manufacturer's protocol, then the cells were incubated for 72 h to ensure knockdown of protein expression. Efficient knockdown of protein expression was confirmed by Western blot (Figs. 4A and S8). Antibodies used: TMED10 (Proteintech 15199-1-AP), TMED2 (Santa Cruz sc-376459), GAPDH (Abcam ab8245).

After 72 h of siRNA incubation, cells were seeded at 50% confluency in glass-bottom plates treated with poly-L-lysine. After a further 24 h, cells were transfected with RUSH construct (LAT, LAT-allL or GPI-EGFP) using Lipofectamine 3000. Four to six hours after transfection, cells were washed with PBS, then incubated with serum-free medium overnight. To synchronize cell cycles, 1 h before biotin addition, cells were treated with full-serum medium. Cells were treated with 100 µM biotin (Sigma-Aldrich) to release cargo proteins.

### Isolation of GPMVs and quantification of raft partition coefficient ($K_{p,raft}$)

Cell membranes were stained with 5 µg/ml of FAST-DiO (Invitrogen), a green fluorescent lipid dye that strongly partitions to disordered phases[20]. After staining, GPMVs were isolated from transfected HeLa cells as described[18]. Briefly, GPMV formation was induced by 2 mM N-ethylmaleimide (NEM) in mildly hypotonic buffer (100 mM NaCl, 10 mM HEPES, 2 mM CaCl2, pH 7.4). To quantify protein partitioning, GPMVs were observed on an inverted epifluorescence microscope (Nikon) at 4 °C after treatment with 200 µM DCA to stabilize phase separation; this treatment has been previously demonstrated not to affect raft affinity of various proteins[59]. The partition coefficient ($K_{p,raft}$) for each protein construct was calculated by dividing the fluorescence intensity of the construct in the raft versus nonraft phase (after background subtraction) for >10 vesicles/trial (e.g., Fig. 1), with >3 independent experiments for each construct.

### Colocalization and spatial distribution quantification

Coloc2 and Mosaic ImageJ plug-ins were used to quantify colocalization and potential interactions, respectively. Coloc2 measurements were used to calculate Pearson's coefficient. For Mosaic, we followed a previously described workflow[60] whose throughput is shown as Supplementary Fig. 3.

### Statistics and reproducibility

Statistical tests were performed on all data sets. The specific test used to test for statistical significance is stated in the individual figure legends.

### Reporting summary

Further information on research design is available in the Nature Portfolio Reporting Summary linked to this article.

## Data availability

All data (including uncropped Western blots) are included in the Supplementary Information or available from the authors, as are reagents generated in this study. The raw numbers for charts and graphs are available in the Source data file whenever possible. Source data are provided with this paper.

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

## Acknowledgements

Funding for this work was provided by the NIH (R35GM134949—I.L., R01AI183581—I.L., R21CA300756—K.R.L.) and the Owens Foundation (I.L.). Microscopy was performed at the Advanced Microscopy Facility at the University of Virginia.

## Author contributions

I.C.S., S.D., K.R.L., and I.L. conceptualized the study and designed the experiments. I.C.S., S.D., and R.I. performed the experiments. I.C.S., S.D., K.R.L., and I.L. analyzed and interpreted the data. I.C.S., K.R.L., and I.L. wrote the manuscript. S.D., K.R.L., and I.L. contributed to the final revised version of the manuscript.

## Competing interests

The authors declare no competing interests.
