## [Transparent Peer Review file · Nature Communications]

ER exit sites mediated by the COPII adaptor sec24D selectively recruit lipid raft-preferring proteins for rapid ER export

Corresponding Author: Professor Ilya Levental

Version 0:

Reviewer comments:

Reviewer #1

(Remarks to the Author)

The paper by Castello-Serrano et al. provides a concise and compact story that largely refines previous scattered work related to ERES function but without providing deeper mechanistic insights into the underlying phenomenology. Prior work has shown ERESs are enriched in cholesterol relative to ER and that Sec24D-associated ERESs are selectively responsible for ER exit of raft-preferring GPI-anchored proteins. Castello-Serrano et al basically repeat these observations and then show that in contrast to Sec24D ERESs, Sec24A ERESs preferentially traffic non-raft preferring membrane cargos. The authors go on demonstrate that Sec24D-associated ERESs are distributed more perinuclear then Sec24A-associated ERESs, with no further mechanistic experiments to understand why or how. They further show that the raft protein LAT is exported from the ER faster than a nonraft LAT variant, again supporting prior work that showed raft GPI proteins exit the ER faster than nonraft proteins and not providing a mechanistic explanation for why or how. Despite its limited novel conclusions, the work is overall well done and should be appreciated by the field of membrane trafficking. Below are comments the authors need to address should the paper be allowed a revision.

Major points

- 1) The authors use the phrase lipid raft-like to describe ERESs, which they confirm are cholesterol-rich relative to ER. Normally, raft-like membranes contain cholesterol and sphingolipids. Do the ERESs the authors show have cholesterol also contain SLs? This should be tested with an SL probe. If not, why are they being called raft-like?
- 2) The fact that the TF-cholesterol probe differentially label ERESs, Golgi and PM depending on the length of incubation with the probe is concerning. Is it possible the probe is perturbing cholesterol trafficking in cells? If so, this could impact all of the observations related to RUSH trafficking and Sec24A/D enrichment or exclusions.
- 3) The authors assume pre-existing ERESs have Sec24D or Sec24A and that cargo enters one or the other ERES selectively based on the cargo's lipid preference. However, the cargo itself could impact the abundance and/or localization of these coat proteins. To help clarify this, the authors should co-express raft and non-raft cargos in the same cell and look for their differential sorting into Sec24A or Sec24D ERESs. All the experiments in the paper are done looking at cells that are individually expressing the cargo constructs without them co-expressed to see real sorting to ERESs.
- 4) Related to Figure 5, there are many indirect explanations for why raft lipid inhibitors myriocin and zaragozic acid could disrupt ER export that don't relate to their impact selectively on ERES lipid composition. Both inhibitors will profoundly affect PM and Golgi lipid composition and circulation, which could indirectly affect ERES function. For this reason, the interpretation of the results in Figure 5 are wide-open.
- 5) Can the authors test whether if they release a non-raft or raft cargo into the secretory pathway, does the ERES that each cargo type goes into become cholesterol enriched?
- 6) Can the authors better describe how they did the quantification of LAT versus LAT-all trafficking to the PM in Figure 1? How do they rule out that LAT-all is endocytosed more rapidly from the PM than LAT and this explains its apparent slower accumulation on the PM? This analysis is important and nontrivial to do.

Reviewer #2

(Remarks to the Author)

The manuscript by Castello-Serrano et al. analyzes the mechanism by which membrane proteins exit from ER. Using the RUSH system, live imaging, and immunostaining, the authors describe that lipid-driven membrane nanodomains known as lipid rafts, play a critical role in ER exit sites. Specifically, raft-associated proteins localize to sec24C and sec24D ER exit sites (ERES), while non-raft proteins localize to sec24A-positive sites. Notably, sec24D, but not sec24A, ERES accumulate a cholesterol analog, suggesting that association with raft-like domains can impact protein export from the ER.

The results clearly demonstrate that lipid interactions play a role in recruiting cargo to ERES. The corresponding author is a leading scientist in the lipid raft field and as with previous excellent papers, the manuscript is well-organized and elegantly written.

However, several important points still need to be addressed, some of which may require additional experiments.

1) The authors describe that sec24C and sec24D preferentially recruit raft-preferring proteins. They suggest that selective "lipid-driven domains" can participate in organizing cargo and trafficking machinery in ERES (as mentioned in the last paragraph of Discussion). However, given that the cholesterol level in ER is known to be less than 5 mol%, it is unlikely that raftophilic lipids can form a liquid-ordered-like phase in the ER. Fig. 1A shows GPMVs derived from the plasma membrane of HeLa cells, not from the ER membrane. This reviewer believes there is a possibility that protein-protein interactions induce the clustering of raftophilic transmembrane proteins, which could result in the enrichment of the cholesterol probe in sec24D patches. For example, if sec24D interacts with certain raftophilic transmembrane proteins, this could lead to the accumulation of raftophilic lipids. In such a scenario, unless a specific protein-protein interaction exists, raftophilic lipids would not be expected to accumulate in large domains. The authors should explicitly rule out the possibility of protein-protein-driven lipid accumulation.

2) Regarding the previous point, it has been reported that ER-to-Golgi transport of the human GPI-anchored protein CD59 requires sec24, with a preference for the sec24C and sec24D isoforms. Additionally, the recycling transmembrane protein complex p24-p23 also shows the same preference for sec24C-sec24D isoforms in ER export (Bonnon et al., *J. Cell Sci.*, 123, 1705-1715, 2010). Co-immunoprecipitation experiments have revealed an unexpected physical interaction between CD59 as well as a GFP-folate-receptor-GPI-anchor hybrid and the p24-p23 complex. It is noteworthy that the p24-p23 complex is a transmembrane protein that interacts with sec24C and sec24D. This reviewer speculates that the p24-p23 complex may accumulate raftophilic lipids, which could then recruit raft-associated cargo proteins such as LAT and GPI-anchored proteins, as shown in the study by Bonnon et al. The authors should further investigate the role of p23-p24 in protein export from the ER, and should discuss the difference between their findings and those of Bonnon et al.

3) In relation to the previous point, Fig. 3D shows a schematic diagram of sec24 isoforms recruiting cargo based on their raft preference. In the figure, sec24A and sec24D are directly associated with non-raft and raft transmembrane proteins, respectively. Although the authors showed colocalization between these proteins, direct interactions are not shown. Considering the findings from Bonnon et al., it is noteworthy that GPI-anchored proteins interact with the p23-p24 complex. Furthermore, another research group reported that p24 family proteins recognize remodeled GPI-anchored proteins and sort them into COPII vesicles (Fujita et al., *J. Cell Biol.*, 194, 61-75, 2011). Specifically, the p24 family proteins interact with GPI-anchored proteins and sec24C/D at the N-terminus and C-terminus, respectively (Kinoshita et al., *Biochim. Biophys. Acta*, 1833, 2473-2478, 2013). The authors should examine whether GPI-anchored proteins are enriched in ERES due to the direct interaction of GPI-anchored proteins with the p24 family protein.

4) The authors employed immunostaining with primary and fluorescently labeled secondary antibodies to visualize the localization of sec24A-D. Additionally, they fused GFP with LAT and its mutant. Prior to immunostaining, the cells were fixed with 4% paraformaldehyde for 10 min at 23 °C. However, the fixation may not be sufficiently strong. If membrane proteins remain mobile, the staining process involving primary and secondary antibodies could potentially induce the clustering of sec24A-D. To mitigate this concern, the authors should compare the distribution of sec24A-D as determined by immunostaining with the distribution observed when sec24A-D is tagged with a fluorescent protein.

Reviewer #3

(Remarks to the Author)

In this publication Castello-Serrano and colleagues make extensive use of the RUSH system to show that transmembrane proteins with preference for ordered or disordered phases are exported from different ER exit sites. For this they use the TM protein linker for activation of T-cells (LAT) as a model. They show that LAT localise to lipid ordered domains while a version of LAT where the TM domain has been replaced with 22 Leucines (LAT-allL) localizes to lipid disordered domains (non-raft). Using RUSH they show that LAT-allL traffics slowly and has higher residency times in the ER. Very interestingly, they show that two isoforms of the COPII coat Sec24A and D localise to segregated ERES, which are enriched in cholesterol and that Sec24D preferentially localises to ERES where the raft-preferring cargo is exported. On the other hand, Sec24A localises cholesterol-depleted ERES where the non-raft cargo is exported. Raft inhibition affects the traffic of LAT but not LAT-allL.

I have really enjoyed reading the paper which is well written and I believe the concept of raft-dependent ER export is novel and would be of great interest for the cell biology community. I have a couple of major points that I would like the authors to address but I would be otherwise very happy to see this published at Nature comms as the model is compelling.

- Could it be possible that LAT-aLL is unfolded or ER export kinetics are slower? This could explain why more of it remains in the ER and why it traffics slowly. The fact that when making GPMV from cells the two proteins localize to different domains only tells about the fraction that localize to the PM at steady state. Inhibition of raft formation may not affect LAT-aLL as its rates of export are so slow anyways.

- If the authors want to claim that Sec24A and D define different classes of ERES they need to prove that both antibodies are indeed labelling ERES by using a COPII subunit that would label all ERES (Sec31, Sec13, etc). One has to be 100% sure the punctae defined by the Sec24A antibodies are not due to unspecific staining.

Minor points

- The authors should provide a more detailed description of their image analysis

- Could the authors explain why they look at ERES at 10 mins after biotin addition (Fig. 3). That's a very late time point considering Weigel at al Figure S2 for example shows that single ERES are emptied out within a minute.

- Fig. 4: Many magenta punctae have green underneath. The authors should not rely on a single line profile to claim that cholesterol is not associated to Sec24A punctae.

- Fig. 5: It is unclear if and how cells before and after biotin addition were quantified (why did the authors did not follow the same cell?). 3 independent experiments with how many cells? In that case single cells should be shown as single points in graphs.

Version 1:

Reviewer comments:

Reviewer #1

(Remarks to the Author)

The authors have addressed all my concerns.

Reviewer #2

(Remarks to the Author)

The authors have addressed my specific concerns and questions from the first round of review by performing more experiments and adding text and figures.

Reviewer #3

(Remarks to the Author)

The authors have addressed all my initial concerns and have also further added mechanistic insights with addition of the TMED figure. I am happy to support publication.

We thank you sincerely for both the positive evaluations and constructive criticisms of our manuscript titled “*ER exit sites mediated by sec24D selectively recruit raft-preferring proteins for rapid ER export*” (NCOMMS-24-27676). We have performed extensive experiments to address all issues raised by the reviewers.

We performed several control experiments that confirmed and extended our conclusions: (a) confirmed the differential sorting of raft versus non-raft cargoes into different ERESs, even in cells co-expressing these cargoes (Supp Fig 6), (b) confirmed enrichment of TF-chol in raft cargo-containing ERES (Fig 5E-H), (c) confirmed that the antibodies for sec24A and sec24D indeed define distinct ERES by verifying their colocalization with fluorescently tagged constructs (Supp Fig 5) and with general antibodies against COPII (Fig 2D-F).

Most significantly, we have added mechanistic insight into our observations by implicating the adaptor proteins of the p24 family in ERES sorting and ER efflux of raft-preferring proteins. Specifically, we found that knock-down of TMED2 (p24b1) and TMED10 (p24d1) suppressed ER efflux of both GPI-anchored protein and LAT, coincident with reducing these cargoes localization to sec24D ERES (Fig 4).

Altogether, we have added 2 main Figures and 2 Supplementary Figures, in addition to significant revision of the manuscript text. These results have confirmed our conclusions that ERES defined by sec24A versus sec24D select cargo based on their lipid raft affinity rather than their cytosolic and luminal features, with sec24D preferentially recruiting raft-preferring proteins and vice versa. The reviewers’ comments and our associated revisions have significantly improved the manuscript by providing more insights on the features of ERESs and the role of p24 family adaptor proteins in these effects. We hope that with these changes you will find the revised manuscript acceptable for publication in *Nature Communications*.

Below is a point-by-point response to all reviewer comments.

Reviewer #1

The paper by Castello-Serrano et al. provides a concise and compact story that largely refines previous scattered work related to ERES function but without providing deeper mechanistic insights into the underlying phenomenology. Prior work has shown ERESs are enriched in cholesterol relative to ER and that Sec24D-associated ERESs are selectively responsible for ER exit of raft-preferring GPI-anchored proteins. Castello-Serrano et al basically repeat these observations and then show that in contrast to Sec24D ERESs, Sec24A ERESs preferentially traffic non-raft preferring membrane cargos. The authors go on demonstrate that Sec24D-associated ERESs are distributed more perinuclear then Sec24A-associated ERESs, with no further mechanistic experiments to understand why or how. They further show that the raft protein LAT is exported from the ER faster than a nonraft LAT variant, again supporting prior work that showed raft GPI proteins exit the ER faster than nonraft proteins and not providing a mechanistic explanation for why or how. Despite its limited novel conclusions, the work is overall well done and should be appreciated by the field of membrane trafficking. Below are comments the authors need to address should the paper be allowed a revision.

We are grateful to the reviewer for their careful review, positive feedback, and helpful suggestion for improving the mechanistic aspects of our paper. We believe the new experiments implicating the TMED2/10 adaptors in sorting of raft cargo to sec24D ERES add important mechanistic insight.

R1-1: The authors use the phrase lipid raft-like to describe ERESs, which they confirm are cholesterol-rich relative to ER. Normally, raft-like membranes contain cholesterol and sphingolipids. Do the ERESs the authors show have cholesterol also contain SLs? This should be tested with an SL probe. If not, why are they being called raft-like?

A1-1: We thank the reviewer for raising this important point. We have limited our usage of “raft” and “raft-like” to proteins and lipids that we (or others) have shown to associate with the raft-like ordered phase of Giant Plasma Membrane Vesicles.

We do believe that the accumulation of these raft-preferring components in specific ERES (marked by sec24D) suggests that these ERES may have some raft-like characteristics, but we carefully revised the manuscript to clearly delineate observations from inferences.

With respect to the raft-like nature of the ERES, lipid rafts are often described as membrane microdomains enriched in both cholesterol and sphingolipids, which interact to form a relatively tightly packed lipid environment. However, complex sphingolipids such as sphingomyelin and glycosphingolipids are essentially absent from the ER under normal conditions, being synthesized in the Golgi and enriched at the plasma membrane¹. Thus, while our data demonstrate that sec24D ERESs are enriched in cholesterol relative to the surrounding ER membrane (Fig 5A,C), we have not observed or tested for sphingolipid enrichment. We do also observe enrichment of raft-associated TMDs at sec24D ERES (Fig 3).

We define rafts as being ‘lipid-driven nanodomains’ arising from preferential interactions between certain lipids. While the classical literature in cells focuses on cholesterol and sphingolipids, in model membranes saturated phospholipids interact with cholesterol similarly to sphingolipids and form liquid-ordered phases to a similar extent^{2,3}. Thus, while sphingolipids facilitate raft-like domains, they are not required.

The specific lipid-lipid and lipid-protein interactions that drive membrane organization in the ER are beyond the scope of our article. Generally, we use the term “raft-like” rather than “raft” to avoid confusing with the strict definitions used in the literature on PM domains.

R1-2: The fact that the TF-cholesterol probe differentially label ERESs, Golgi and PM depending on the length of incubation with the probe is concerning. Is it possible the probe is perturbing cholesterol trafficking in cells? If so, this could impact all of the observations related to RUSH trafficking and Sec24A/D enrichment or exclusions.

A1-2: This is a good point, we are also somewhat confused and concerned by the time/concentration dependence of labeling with TF-cholesterol. However, this is not an issue for the vast majority of the experiments we reported. TF-cholesterol was not used in any of the RUSH assays, nor the colocalization experiments between Sec24 and TMD probes. The only experiments using TF-Chol are those shown in Figure 4, showing colocalization between ER-associated TF-chol puncta and Sec24D, but not Sec24A. We have also added new experiments showing that the TF-Chol puncta colocalize with raft-preferring LAT, but not with non-raft LAT-allL (Fig 5E-H).

Thus, despite the caveats of the TF-Chol labeling, its specificity for sec24D and LAT-enriched puncta in the ER is notable and evidences their raft-like character.

Additionally, we have included a Discussion point about the caveats with TF-chol:

“The enrichment of a fluorescent cholesterol analog (TF-cholesterol) in ER structures (Fig 5) has been previously reported, but is somewhat surprising, since the majority of native cellular cholesterol is in the plasma membrane and endosomes. First, it is important to emphasize that fluorescent lipids are imperfect analogs of their native counterparts, though TF-Chol retains the raft phase affinity of native cholesterol better than most. We analyzed the staining at various concentrations and time points, noting that while some conditions efficiently stained ER structures, lower concentrations and shorter times showed no ER staining but rather plasma membrane accumulation (Supp Fig 6). We suspect that these distributions reflect aspects of subcellular cholesterol traffic, especially under cholesterol over-loading, which may allow visualization of cholesterol-rich ERES structures that would not be observable under steady-state conditions.”

R1-3: The authors assume pre-existing ERESs have Sec24D or Sec24A and that cargo enters one or the other ERES selectively based on the cargo’s lipid preference. However, the cargo itself could impact the abundance and/or localization of these coat proteins. To help clarify this, the authors should co-express raft and non-raft cargos in the same cell and look for their differential sorting into Sec24A or Sec24D ERESs. All the experiments in the paper are done looking at cells that are individually expressing the cargo constructs without them co-expressed to see real sorting to ERESs.

A1-3: We thank the reviewer for this point and clear suggestion: we have performed exactly the co-expression experiments suggested, expressing the raft-preferring (LAT) and non-raft-preferring (LATallL) cargos simultaneously in the same cell. Consistent with our previous report, under these co-expression conditions, raft-preferring cargo (LAT) localized preferentially to sec24D-positive ERESs, while non-raft cargo (LATallL) predominantly associated with Sec24A-positive ERESs (see Supp Figure 6, reproduced below). These observations strengthen the conclusion that cargo sorting is guided by pre-existing specialization of ERES subtypes, rather than cargo-induced recruitment of specific Sec24 isoforms.

Supplementary Figure 6. sec24 isoforms colocalize with cargo proteins based on their raft affinity. Quantification of LAT or LAT-allL puncta observable 10 minutes of biotin treatment to release ER RUSH with puncta of either (A) sec24D or (B) sec24A. Empty symbols correspond to individual cells, solid symbols to means of 3 individual experiments with >10 cells each.

R1-4: Related to Figure 5, there are many indirect explanations for why raft lipid inhibitors myriocin and zaragozic acid could disrupt ER export that don't relate to their impact selectively on ERES lipid composition. Both inhibitors will profoundly affect PM and Golgi lipid composition and circulation, which could indirectly affect ERES function. For this reason, the interpretation of the results in Figure 5 are wide-open.

A1-4: This is a fair caveat; however, we point out that the specificity for sec24D versus sec24A ERES (Fig 6A-B) and for raft versus non-raft cargo kinetics (Fig 6C-E) support the interpretation that these effects are associated with raft-like membrane organization and raft-dependent trafficking. Further, we did not specifically link our observations to ERES lipid composition, rather leaving the interpretation open, as the reviewer described. We have included a note to this point the revision:

"It is important to note that the effects of these inhibitors are not restricted to the ER, as both cholesterol and sphingolipids distribute broadly and enrich in the plasma membrane after their synthesis, thus our observations do not necessarily implicate changes in ERES lipid composition."

R1-5: Can the authors test whether if they release a non-raft or raft cargo into the secretory pathway, does the ERES that each cargo type goes into become cholesterol enriched?

A1-5: This was a good suggestion. We tested this performing RUSH-release experiments for raft-preferring or non-raft cargo and assessing cholesterol enrichment at the corresponding ERESs using a fluorescent cholesterol probe (TF-cholesterol). Consistent with our conclusions, we found that ERESs accumulating raft-preferring cargo (LAT) indeed enrich for cholesterol, whereas ERESs accumulating non-raft cargo do not. These data are now included as Fig 5E-H and discussed in Results and Discussion sections.

R1-6: Can the authors better describe how they did the quantification of LAT versus LAT-allL trafficking to the PM in Figure 1? How do they rule out that LAT-allL is endocytosed more rapidly from the PM than LAT and this explains its apparent slower accumulation on the PM? This analysis is important and nontrivial to do.

A1-6: We apologize for the lack of clarity in our methodological explanation. Figure 1D-E shows the time course of ER efflux for the two constructs. These data were obtained by confocal live-imaging of cells expressing LAT or LAT-allL tagged with fluorescent proteins (mRFP or EGFP). At each time point, the fluorescent signal in the ER was measured by applying a mask generated from the 0 time point (ie without biotin), at which all signal for all constructs is in the ER. This fluorescence in the ER was then divided by the fluorescence in the cell as a whole to calculate the fraction of target protein in the ER.

This “fraction remaining in the ER” was then quantified in images taken every 5 minutes after biotin introduction. Data points in Fig 1D show the protein fraction remaining in the ER over the 90-minute time course, showing averages +/- standard deviation of 5 cells. Because this analysis focus on the early time points of post-ER trafficking, endocytosis and Golgi-to-PM trafficking are not relevant.

These details have now been clarified in the Figure 1 legend and in the Methods section. We also expanded the other sections describing the imaging and quantifications.

Reviewer #2

The manuscript by Castello-Serrano et al. analyzes the mechanism by which membrane proteins exit from ER. Using the RUSH system, live imaging, and immunostaining, the authors describe that lipid-driven membrane nanodomains known as lipid rafts, play a critical role in ER exit sites. Specifically, raft-associated proteins localize to sec24C and sec24D ER exit sites (ERES), while non-raft proteins localize to sec24A-positive sites. Notably, sec24D, but not sec24A, ERES accumulate a cholesterol analog, suggesting that association with raft-like domains can impact protein export from the ER. The results clearly demonstrate that lipid interactions play a role in recruiting cargo to ERES. The corresponding author is a leading scientist in the lipid raft field and as with previous excellent papers, the manuscript is well-organized and elegantly written.

We thank the reviewer for their careful analysis and are humbled by their generous feedback.

R2-1: The authors describe that sec24C and sec24D preferentially recruit raft-preferring proteins. They suggest that selective “lipid-driven domains” can participate in organizing cargo and trafficking machinery in ERES (as mentioned in the last paragraph of Discussion). However, given that the cholesterol level in ER is known to be less than 5 mol%, it is unlikely that raftophilic lipids can form a liquid-ordered-like phase in the ER. Fig. 1A shows GPMVs derived from the plasma membrane of HeLa cells, not from the ER membrane. This reviewer believes there is a possibility that protein-protein interactions induce the clustering of raftophilic transmembrane proteins, which could result in the enrichment of the cholesterol probe in sec24D patches. For example, if sec24D interacts with certain raftophilic transmembrane proteins, this could lead to the accumulation of raftophilic lipids. In such a scenario, unless a specific protein-protein interaction exists, raftophilic lipids would not be expected to accumulate in large domains. The authors should explicitly rule out the possibility of protein-protein-driven lipid accumulation.

A2-1: Thanks for this thoughtful input. Overall, we agree: ER cholesterol concentration seems too low for liquid-ordered domain formation, and while the levels of saturated lipids are unknown, the low sphingolipid concentration also suggests that the bulk ER lipid composition is not optimized for phase separation. We also agree that adaptor proteins and their multivalent protein-protein interactions likely play a critical role in amplifying the lipid-lipid interactions associated with raft domains.

We have amended our Discussion significantly to emphasize these points and avoid the impression that lipid-lipid interactions are solely, or even primarily, responsible for the effects we describe. We hope the Reviewer does not object to us using some of their wording in our manuscript:

“These observations imply a role for lipid-mediated domains in the organization and function of the early secretory system. This inference is surprising in light of the ER’s relatively low levels of cholesterol⁴⁻⁶, which is essential for formation of ordered lipid domains^{7,8}. Similarly, complex sphingolipids – including sphingomyelin and glycosphingolipids – are synthesized in the Golgi and therefore unlikely to be present at significant abundances in the ER⁹. However, membranes with cholesterol concentrations at or even below 10 mol% can phase separate under certain conditions^{2,3}, even without considering the possible influence of local differences in lipid compositions (e.g. due to local production or lateral confinements) or the effects of protein assembly on membrane organization¹⁰. Indeed, such multimeric protein assemblies underlie the structure of ERES, suggesting the possibility that protein-protein interactions induce the clustering of raft-preferring transmembrane proteins, which in turn results in recruitment of raft-preferring lipids like cholesterol. Such effects may be responsible for the cholesterol-enriched ER structures that were recently reported¹¹ and confirmed here (Fig 5).”

R2-2&3: Regarding the previous point, it has been reported that ER-to-Golgi transport of the human GPI-anchored protein CD59 requires sec24, with a preference for the sec24C and sec24D isoforms. Additionally, the recycling transmembrane protein complex p24-p23 also shows the same preference for sec24C-sec24D isoforms in ER export (Bonnon et al., J. Cell Sci., 123, 1705-1715, 2010). Co-immunoprecipitation experiments have revealed an unexpected physical interaction

between CD59 as well as a GFP-folate-receptor-GPI-anchor hybrid and the p24-p23 complex. It is noteworthy that the p24-p23 complex is a transmembrane protein that interacts with sec24C and sec24D. This reviewer speculates that the p24-p23 complex may accumulate raftophilic lipids, which could then recruit raft-associated cargo proteins such as LAT and GPI-anchored proteins, as shown in the study by Bonnon et al. The authors should further investigate the role of p23-p24 in protein export from the ER, and should discuss the difference between their findings and those of Bonnon et al.

...considering the findings from Bonnon et al., it is noteworthy that GPI-anchored proteins interact with the p23-p24 complex. Furthermore, another research group reported that p24 family proteins recognize remodeled GPI-anchored proteins and sort them into COPII vesicles (Fujita et al., *J. Cell Biol.*, 194, 61-75, 2011). Specifically, the p24 family proteins interact with GPI-anchored proteins and sec24C/D at the N-terminus and C-terminus, respectively (Kinoshita et al., *Biochim. Biophys. Acta*, 1833, 2473-2478, 2013). The authors should examine whether GPI-anchored proteins are enriched in ERES due to the direct interaction of GPI-anchored proteins with the p24 family protein.

A2-3&4: This was an excellent suggestion that prompted us to explore the role of the p24-p23 proteins in our system. Specifically, the p24 family members TMED2 (p24b1) and TMED10 (p24d1) have been implicated by Bonnon et al¹² and others¹³ in GPI-AP trafficking potentially associated with lipid domains¹⁴, so we pursued knockdown experiments to determine their involvement in LAT ER efflux and COPII localization. First, we confirmed previous observations of their effects on GPI-AP trafficking, showing that dual knockdown of TMED2/TMED10 (by siRNA) dramatically suppressed ER efflux of GPI-anchored GFP. While ~50% of GPI-GFP trafficked to the Golgi 15 min after release of RUSH by biotin, less than 20% reached the Golgi with the TMED2/10 KD (Fig 4). We also observed that TMED2/10 KD induced a notable delay in colocalization between GPI-GFP and sec24D (Fig 4). These observations are fully consistent with the previously reported role of TMED2/10 in recruiting GPI-APs to raft-like sec24D ERES for ER efflux^{12,14}.

Next, we examined the role of TMED2/10 on the trafficking of LAT and LAT-all: as for GPI-GFP, TMED2/10 siRNA significantly reduced ER-to-Golgi transport of LAT after RUSH release (Fig 4). And as for GPI-GFP, this effect coincided with attenuated colocalization of LAT with sec24D. Both effects were specific for the raft probe, as neither ER efflux nor sec24D colocalization of LAT-all were affected by the knockdowns.

These observations are consistent with the previously hypothesized role of TMED2/10 as adaptors for recruiting raft-associated cargo to sec24C/D ERES to facilitate their secretory trafficking. The specific protein-protein and protein-lipid interactions underlying these effects remain to be elucidated and are beyond the scope of the present work. Because LAT is a transmembrane protein, sec24 could in principle directly interact with cytosolic sorting motifs; however, the specificity of sec24 isoforms for either raft (sec24D for LAT) or non-raft (sec24A for LAT-all) TMDs on the same protein suggests other important interactions. Another possibility is that TMED2/10 proteins recruit GPI-APs and perhaps other raft-associated cargo, which nucleates or assembles a raft-like domain, to which LAT is preferentially recruited. These exciting hypotheses will comprise the direction of our future work.

These findings have been added as Fig 4, with an extended addition to the Results and Discussion sections.

Finally, because we do not have evidence for direct interactions between cargo, sec24 isoforms, and TMED2/10, we removed the schematic cartoon (previous Fig 3D).

R2-5: The authors employed immunostaining with primary and fluorescently labeled secondary antibodies to visualize the localization of sec24A-D. Additionally, they fused GFP with LAT and its mutant. Prior to immunostaining, the cells were fixed with 4% paraformaldehyde for 10 min at 23 °C. However, the fixation may not be sufficiently strong. If membrane proteins remain mobile, the staining process involving primary and secondary antibodies could potentially induce the clustering of sec24A-D. To mitigate this concern, the authors should compare the distribution of sec24A-D as determined by immunostaining with the distribution observed when sec24A-D is tagged with a fluorescent protein.

A2-5: Good point; we did this control and observed clear colocalization between immunostained and FP-tagged sec24A and also for sec24D, but not the non-matching antibody (i.e. minimal colocalization between anti-sec24A and mCherry-sec24D). Further, the bright puncta of both sec24 isoforms were observable without fixation / immunostaining. Thus, while we cannot be sure that immunostaining reveals exactly the full localization of sec24 isoforms, their organization into distinct puncta is not an artifact. These data are included as Supplementary Figure 5.

Supplementary Figure 5. Confirming the specificity of anti-sec24 antibodies in cells transfected with sec24A/D. Representative images of cells transfected with sec24 isoforms fused to fluorescent proteins and subsequent staining with antibodies against either the same isoform or the “opposite” one. Staining shows that the antibodies distinguish the distinct isoforms, which form distinct puncta. Black in the merge channel represents colocalization.

Reviewer #3

In this publication Castello-Serrano and colleagues make extensive use of the RUSH system to show that transmembrane proteins with preference for ordered or disordered phases are exported from different ER exit sites. For this they use the TM protein linker for activation of T-cells (LAT) as a model. They show that LAT localise to lipid ordered domains while a version of LAT where the TM domain has been replaced with 22 Leucines (LAT-allL) localizes to lipid disordered domains (non-raft). Using RUSH they show that LAT-allL traffics slowly and has higher residency times in the ER. Very interestingly, they show that two isoforms of the COPII coat Sec24A and D localise to segregated ERES, which are enriched in cholesterol and that Sec24D preferentially localises to ERES where the raft-preferring cargo is exported. On the other hand, Sec24A localises cholesterol-depleted ERES where the non-raft cargo is exported. Raft inhibition affects the traffic of LAT but not LAT-allL. I have really enjoyed reading the paper which is well-written and I believe the concept of raft-dependent ER export is novel and would be of great interest for the cell biology community.

We thank the reviewer for their careful analysis and generous feedback.

R3-1: Could it be possible that LAT-allL is unfolded or ER export kinetics are slower? This could explain why more of it remains in the ER and why it traffics slowly. The fact that when making GPMV from cells the two proteins localize to different domains only tells about the fraction that localize to the PM at steady state. Inhibition of raft formation may not affect LAT-allL as its rates of export are so slow anyways.

A3-1: We believe folding of LAT-allL is unlikely to be an important contributor to our observations for several reasons: (a) LAT is an unusual membrane protein that has essentially no stable folding: its extracellular portion is only three amino acids, followed by a single-pass TMD whose proper folding is evidenced by the fact that it remains in membranes rather than being soluble, and its intracellular portion is predicted to be completely disordered, (b) the RUSH assay accumulates proteins in the ER for many hours, thus even slowly folding proteins would have a chance to fold, while mis-folded proteins would have ample opportunity to be removed.

R3-2: If the authors want to claim that Sec24A and D define different classes of ERES they need to prove that both antibodies are indeed labelling ERES by using a COPII subunit that would label all ERES (Sec31, Sec13, etc). One has to be 100% sure the punctae defined by the Sec24A antibodies are not due to unspecific staining.

A3-2: This is a good point. To confirm that sec24A and sec24D indeed define distinct ERES, we performed two distinct experiments:

First, we tested the isoform specificity and accuracy of the sec24 antibodies by immunostaining cells expressing sec24 isoforms tagged with fluorescent proteins. We observed that anti-sec24A stained the same puncta observed with sec24A-mCherry while anti-sec24D colocalized with sec24D-mCherry (but not *vice versa*).

Second, we performed the experiment suggested by the reviewer: staining with a polyclonal COPII antibody (PA1-069A) and analyzing its colocalization with sec24A and sec24D puncta. Indeed, the COPII antibody stained most puncta that were either sec24A or sec24D positive, but the two sec24 isoform antibodies rarely stained the same puncta.

Both data sets confirm that these two sec24 isoforms assemble into distinct puncta and are included as Supplementary Figure 5 (and above in A2-5) and Fig 2D-F, respectively.

R3-3: The authors should provide a more detailed description of their image analysis.

A3-3: We apologize for the lack of clarity in our methodological explanation. We have added the following sections to the Methods sections:

Imaging and quantification: Unless specified, imaging was performed on a confocal microscope using three channel acquisition (green = ex:488/em:509, red=ex:584/em:609, far-red=ex:584/em:610) in a sequential manner. When possible, cells were imaged live; for immunofluorescence, cells were fixed with 4% paraformaldehyde (PFA) for 10 min at 23°C. anti-58k (1:500 mouse, Abcam ab27043) was used as cis-Golgi marker. Antibodies against sec24 isoforms were Sigma-Aldrich sec24A HPA038901, sec24B HPA038181, sec24C HPA040196, sec24D HPA053486, sec24A Santa Cruz sc-517155, sec24D ThermoFisher 67409-1-IG. The polyclonal COPII antibody was ThermoFisher PA1-069A.

Immunostaining was performed with secondary antibodies conjugated with Alexa fluorophore dyes. Acquired images were quantified using Fiji. Coloc2 measurements were used to calculate Pearson's coefficient. Puncta colocalization was calculated by identifying puncta using the Fiji thresholding algorithm, then manually counting the LAT/LAT-allL puncta overlapping with sec24A/sec24D puncta by the total number of LAT/LATallL puncta. For Fig 5D, "Fraction of ER-positive cells" was calculated by manually scoring cells with ER signal above background at various time points; for Fig 5E, "ER residence time" was calculated by fitting the curves in Fig 5D to a single-exponential decay and deriving the time at which 50% of cells were ER positive. To calculate the fraction of construct in the Golgi (Fig 6), a Golgi mask was created by immunostaining fixed cells against the Golgi marker anti-58K, then comparing the fluorescence signal from the Golgi mask to the whole cell.

Colocalization and spatial distribution quantification: Coloc2 and Mosaic ImageJ plug-ins were used to quantify colocalization and potential interactions, respectively. Coloc2 measurements were used to calculate Pearson's coefficient. For Mosaic, we followed a previously described workflow¹⁵ whose throughput is shown as Supp Fig 3.

R3-4: Could the authors explain why they look at ERES at 10 mins after biotin addition (Fig. 3). That's a very late time point considering Weigel at al Figure S2 for example shows that single ERES are emptied out within a minute.

A3-4: This is a good question. We observed quite different ER exit kinetics for some of our constructs compared to TNF- α , with LAT and GPI-EGFR having ~2-fold slower ER efflux than TNF- α ^{11,16}. Similar differences in ER exit kinetics were recently reported between TNF- α and CCR5 (a GPCR). These observations echo classical reports that proteins can have greatly varying rates of ER to Golgi transport, with ER residence times of <15 min to >2 hrs¹⁷.

R3-5 Fig. 4: Many magenta punctae have green underneath. The authors should not rely on a single line profile to claim that cholesterol is not associated to Sec24A punctae.

A3-5: This is a fair point – there are many puncta in both TF-chol and sec24A images and some do appear to colocalize.

We have softened this description:

“These TF-chol puncta colocalized strongly with sec24D-positive structures, but much less with sec24A (Fig 5). Intensity profiles of magnified regions show clear overlap of TF-chol with sec24D (Fig 5A), with less correlation with sec24A (Fig 5B). Thus, consistent with the preference of sec24 isoforms for raft-associated proteins, sec24D puncta may be relatively enriched in cholesterol.”

R3-6 Fig. 5: It is unclear if and how cells before and after biotin addition were quantified (why did the authors did not follow the same cell?). 3 independent experiments with how many cells? In that case single cells should be shown as single points in graphs.

A3-6: Apologies for confusion. Data in Figure 6D are not from individual cells being tracked over time, but rather population measurements in which many cells are counted at each time point and scored as either positive or negative for having signal of the construct in the ER above background. Thus, data points represent the fraction of ER-positive cells at a given time point, rather than individual cells. The reason for this slightly inelegant approach, rather than measuring individual cells over time in a live cell experiment, is that quantifying ER residence in an individual cell over 3 hrs is technically difficult.

This detail has now been clarified in the Methods.

- 1 Holthuis, J. C., Pomorski, T., Raggars, R. J., Sprong, H. & Van Meer, G. The organizing potential of sphingolipids in intracellular membrane transport. *Physiol Rev* **81**, 1689-1723 (2001)
- 2 Veatch, S. L. & Keller, S. L. Separation of liquid phases in giant vesicles of ternary mixtures of phospholipids and cholesterol. *Biophys J* **85**, 3074-3083 (2003)
- 3 Veatch, S. L. & Keller, S. L. Miscibility phase diagrams of giant vesicles containing sphingomyelin. *Phys. Rev. Lett.* **94**, 148101 (2005)
- 4 van Meer, G., Voelker, D. R. & Feigenson, G. W. Membrane lipids: where they are and how they behave. *Nat Rev Mol Cell Biol* **9**, 112-124 (2008).2642958
- 5 Lange, Y. Disposition of intracellular cholesterol in human fibroblasts. *J Lipid Res* **32**, 329-339 (1991)
- 6 Radhakrishnan, A., Goldstein, J. L., McDonald, J. G. & Brown, M. S. Switch-like control of SREBP-2 transport triggered by small changes in ER cholesterol: a delicate balance. *Cell Metab* **8**, 512-521 (2008).PMC2652870
- 7 Sengupta, P., Baird, B. & Holowka, D. Lipid rafts, fluid/fluid phase separation, and their relevance to plasma membrane structure and function. *Semin Cell Dev Biol* **18**, 583-590 (2007).2147712
- 8 Levental, I. *et al.* Cholesterol-dependent phase separation in cell-derived giant plasma-membrane vesicles. *Biochem J* **424**, 163-167 (2009)
- 9 Holthuis, J. C. & Menon, A. K. Lipid landscapes and pipelines in membrane homeostasis. *Nature* **510**, 48-57 (2014)
- 10 Wang, H. Y. *et al.* Coupling of protein condensates to ordered lipid domains determines functional membrane organization. *Science advances* **9**, eadf6205 (2023).PMC10132753
- 11 Weigel, A. V. *et al.* ER-to-Golgi protein delivery through an interwoven, tubular network extending from ER. *Cell* **184**, 2412-2429 e2416 (2021)
- 12 Bonnon, C., Wendeler, M. W., Paccaud, J. P. & Hauri, H. P. Selective export of human GPI-anchored proteins from the endoplasmic reticulum. *J Cell Sci* **123**, 1705-1715 (2010)

- 13 Anwar, M. U. *et al.* ER-Golgi-localized proteins TMED2 and TMED10 control the formation of plasma membrane lipid nanodomains. *Dev Cell* **57**, 2334-2346 e2338 (2022)
- 14 Fujita, M. & Kinoshita, T. Structural remodeling of GPI anchors during biosynthesis and after attachment to proteins. *FEBS Lett* **584**, 1670-1677 (2009)
- 15 Shivanandan, A., Radenovic, A. & Sbalzarini, I. F. MosaicIA: an ImageJ/Fiji plugin for spatial pattern and interaction analysis. *BMC bioinformatics* **14**, 349 (2013).PMC4219334
- 16 Castello-Serrano, I. *et al.* Partitioning to ordered membrane domains regulates the kinetics of secretory traffic. *eLife* **12**, RP89306 (2023)
- 17 Fries, E., Gustafsson, L. & Peterson, P. A. Four secretory proteins synthesized by hepatocytes are transported from endoplasmic reticulum to Golgi complex at different rates. *EMBO J* **3**, 147-152 (1984).PMC557311